

# Laboratory evaluation of a bio-insecticide candidate from tangerine peel extracts against *Trialeurodes vaporariorum* (Homoptera: Aleyrodidae)

Nancy Flores[1], Julia Prado[2], Rosario Espin[2], Hortensia Rodríguez[3] and José-Manuel Pais-Chanfrau[2]

[1] Ibarra, Imbabura, Ecuador
[2] FICAYA/Carrera de Agroindustria, Universidad Técnica del Norte (UTN), Ibarra, Imbabura, Ecuador
[3] School of Chemical Sciences and Engineering, Yachay Tech University, Urcuquí, Imbabura, Ecuador

Corresponding author
José-Manuel Pais-Chanfrau,
jmpais@utn.edu.ec

## ABSTRACT

**Background:** The excessive use of synthetic insecticides in modern agriculture has led to environmental contamination and the development of insect resistance. Also, the prolonged use of chemical insecticides in producing flowers and tomatoes in greenhouses has caused health problems for workers and their offspring. In this study, we analyzed the efficacy of mandarin peel (*Citrus reticulata* L.) essential oil (EO) as a natural insecticide against greenhouse whitefly (*Trieurodes vaporariorum* W., Homoptera: Aleyrodidae), a common pest in greenhouse production of different crops.

**Methods:** Petroleum ether (PET) and n-hexane (HEX) were used as solvents to extract essential oil (EO) from tangerine peels.

**Results:** The yield of EO was 1.59% and 2.00% (m/m) for PET and HEX, respectively. Additionally, the insect-killing power of EO was tested by checking how many greenhouse whiteflies died at different times. The results showed that PET and HEX extracts of tangerine EO effectively controlled greenhouse whiteflies. Furthermore, with both solvents, a 12.5% (v/v) application was as practical as the commercial insecticide imidacloprid. Further characterization tests with the polarimeter, FTIR, HPLC-RP, and GC-MS showed that the essential oil (EO) contained about 41% (v/v) of *d*-limonene and that this compound may be responsible for the observed insecticidal properties.

**Conclusion:** Therefore, tangerine peel essential oil is an excellent botanical insecticide candidate for controlling greenhouse whiteflies.

## INTRODUCTION

Modern agriculture was sustained using chemicals for much of the 20th century (*Davydov et al., 2018*). Their use ensured higher crop yields and satisfying the increasing demand for

agricultural products (*Lanz, Dietz & Swanson, 2018*). At a global level, this revolution led to the fact that, for the first time, food production worldwide could satisfy demand (*McLennon et al., 2021*).

However, some researchers pointed out that the extensive and often abusive use of chemical pesticides also caused adverse effects on agro-food ecosystems (*Arora & Sahni, 2016*) through different mechanisms such as rain, irrigation systems and harvest and post-harvest activities. These chemicals contaminate the ecosystem. Additionally, some amounts of these chemical pesticides can directly contaminate agricultural products and potentially endanger human and animal health (*Yang et al., 2022*).

These problems are evident in the case of chemical insecticides, which, by controlling insect pests, often have negative effects on pollinating insects like honeybees that guarantee the robustness and vitality of ecosystems (*Nicolopoulou-Stamati et al., 2016*; *Sharma & Singhvi, 2017*). Chemical biocides (like bactericides, fungicides, and nematicides) have also had similar effects, affecting the beneficial endophytic microflora of crops (*Reshma, Vinaya & Linu, 2019*).

These long-term practices have resulted in the erosion of arable land (*Sabatier et al., 2014*) and the loss of the necessary balance between crop health and the soil microflora (*Anderson, Armstrong & Smith, 1981*). They have also led to the appearance of resistance genes that allow different pests to survive and develop. Such is the case with chemical insecticides. There are reports of the emergence of resistance genes among numerous insect species that have acquired various resistance genes to multiple types of chemical insecticides, forcing farmers to use higher doses of these chemicals and worsening the problem (*Karatolos et al., 2010*; *Riaz et al., 2013*).

Strategies have been devised to mitigate the impact of the extensive use of chemical fertilizers and pesticides, such as integrated pest management, where chemicals are used minimally, alongside other control strategies, such as biologicals and other natural products (*Conboy et al., 2020*). Among the latter are the so-called "natural" pesticides, products commonly obtained from plants (*Smith & Perfetti, 2020*; *Ngegba et al., 2022*).

Furthermore, if the raw material for these botanical pesticides were agricultural or agro-industrial waste, the cost of producing the botanic pesticide with them and the overall environmental impact of the agricultural and agro-industry waste used would become low (*Sindhu et al., 2020*).

Numerous studies have demonstrated the bactericidal, fungicidal, antioxidant, and insecticidal activity of essential oil (EO) extracts from citrus peel (*Kholaf, Gomaa & Ziena, 2017*; *Mahmoud, 2017*; *Sreepian et al., 2019*). These qualities make it suitable for different treatments for various health pathologies (*Anandakumar, Kamaraj & Vanitha, 2021*) such as cancer (*Nair et al., 2018*) and exert specific allelopathic (*El Sawi et al., 2019*) and health-promoting (*Liu et al., 2021*) effects. Furthermore, in agriculture, the prevention of pest damage is crucial during the cultivation of flowers, fruits, and vegetables, as well as during the transportation and storage of agricultural products.

Some natural substances present in fruits and plant tissues have evolved to become repellents for insects and other pests (*Oguh et al., 2019*). However, the mechanism by which this occurs is not fully understood. Such is the case with some terpenes, such as

limonene (*Feng et al., 2020*; *Gültepe, 2020*; *de Brito et al., 2021*). This monoterpene has demonstrated biocidal and insecticidal effects against various pests (*de Brito et al., 2021*). Limonene is found in the peels of many citrus fruits (*John, Muthukumar & Arunagiri, 2017*). The beneficial effects of *d*-limonene, one of the main components of citrus peel essential oils, have also been reported on human and animal health (*Anandakumar, Kamaraj & Vanitha, 2021*).

The tangerine (*Citrus reticulata* L.) is a citrus fruit with a pleasant flavor and is a source of vitamins and minerals (*Khan et al., 2010*). In Ecuador, in the Pimampiro canton, Imbabura province, there are relatively large (more than 400 ha, according to the opinion of a local expert) tangerine plantations.

It is estimated, however, that nearly 20–30% of the fruits are wasted at the peak of the mandarin harvest, as occurs elsewhere (*Caicedo Jiménez, 2021*). The overproduction of tangerines brings unpleasant effects on health and the environment, especially in the areas surrounding municipal landfills. Therefore, finding alternatives to recover these crop wastes is necessary (*Punvichai & Pioch, 2019*; *Panwar, Panesar & Chopra, 2021*). In the case of tangerine, it might be convenient to find economically feasible ways to use the tangerine peel to produce essential oils with high contents of bioactive substances such as *d*-limonene (*Mandal & Mandal, 2016*).

Finally, the northern Andean zone of Ecuador (formed by the provinces of Pichincha, Imbabura, and Carchi), whose economies depend on agriculture, has recently extended the crops protected in greenhouses in recent years (*Knapp, 2019*; *Nazeeh & Suárez-López, 2020*). One pest affecting these crops is the greenhouse whitefly (*Trialeurodes vaporariorum* W. (Homoptera: Aleyrodidae)), which must be adequately controlled (*Rincon et al., 2019*).

Chemical insecticides like neonicotinoids and organophosphorus are frequently used to control this pest. Imidacloprid is among the most widely used and efficient neonicotinoid insecticides to control greenhouse whiteflies.

However, organophosphorus and neonicotinoid insecticides in closed environments, such as greenhouses, have been associated with occupational diseases in greenhouse workers exposed to them (*Zhang et al., 2022*; *Zhao et al., 2022*) and damage to pollinating insects such as bees (*Godfray et al., 2014*).

On the other hand, natural insecticides, botanical insecticides, or bio-insecticides are safe for plants and people (*Ahmed et al., 2022*; *Riyaz et al., 2022*), even though they are often not as effective as chemical insecticides (*Damalas & Koutroubas, 2020*).

It is also recognized that they are less susceptible to the appearance of resistance in the pests they intend to control (*Cardona et al., 2001*). The multiple mechanisms that the active components of bioinsecticides use to exert their effects against insect pests may be the most likely explanation for the latter (*Adisa et al., 2019*; *Tocmo et al., 2020*).

Such is the case for extracts of EO from citrus peels, where the mechanisms by which the EO components act on insects are not fully understood. However, mechanisms such as repelling (*Isman, 2006*), direct action on the external membranes of the insect, which causes dehydration and death (*Feng et al., 2020*), antifeedant activity (*Widjayanti, Tarno &*

*Anggiah, 2018*), or toxic effects on the nervous system of insects (*Karr, 2014*), have been suggested by some authors as possible mechanisms of action of these substances.

This study aims to see what happens to greenhouse whiteflies when different doses of essential oil from tangerine peels are used to determine the potentiality of solvent extract from tangerine peels as a botanical insecticide.

# MATERIALS AND METHODS

## Location and origin of the tangerine fruits used

The tangerine fruits (*Citrus reticulata* L. var. clementine) used in this study originated from the Pimampiro Canton (0°24′0″N, 77°58′12″W), located in the extreme east of the Imbabura province in northern Ecuador. The harvested fruits were in their optimal state of maturity when the fruit peel was easily separated from the rest of the fruit.

Before separating the peels, the tangerine fruits were washed with tap water to remove dust that might have adhered during transportation and dried individually with a dry cloth. After removing the peels, they were cut into 5–10 mm pieces to facilitate the essential oil extraction.

## Solvent extraction at laboratory scale

The extracts were obtained in a 250 mL Soxhlet apparatus, to which 50 g of tangerine peel was added, and petroleum ether or n-hexane was used as the solvent, as reported elsewhere (*Park, Ko & Kim, 2015*). Approximately 4 h after the start of the extraction, it was considered finished (see in Supplemental Files: Data.zip: Apparatus for the extraction and recovery solvents).

After extraction with solvents (n-hexane or petroleum ether), the extracts were placed in a rotary evaporator to remove the solvents from the extracts. The boiling points of pure solvents are between 40–60 °C for petroleum ether and 69 °C in n-hexane, while components such as *d*-limonene have a boiling point of approximately 178 °C, values obtained at sea level. These values are even lower at the height at which the experiments were carried out in this study (~2,200 m above sea level). The temperature of the vapors removed from the solvents in both cases was observed using a thermometer placed in the roto evaporator condenser. The solvent removal process was carried out until it was observed that the temperature of said vapors exceeded 100 °C, so it is to be assumed that in said extracts the amount of existing solvent is almost nil. This last process did not exceed 90 min, in any case.

## Determination of the cumulative mortality of the greenhouse whitefly

Ten self-made entomological boxes (each 30 cm × 30 cm × 48 cm) were built with tulle fabric for whitefly rearing (see in Supplemental Files: Data.zip: Devices for culturing whiteflies and carrying out experiments). Each box contained a healthy tomato (*Solanum lycopersicum* L.) seedling planted in a pot and left for 60 days to grow.

On the other hand, in a tomato greenhouse with a high greenhouse whitefly (*Trialeurodes vaporariorum* W.) infestation, about 200 adult whiteflies were collected with an insect aspirator and released in each entomological box at a rate of about 20 per box.

After 30 days of whitefly breeding, nearly five adult flies were collected from each entomological box through an insect aspirator. Then just ten adult whiteflies were randomly placed inside each 250-mL bottle.

Before, the moistened sterile cotton was placed with a total volume of 177.8 µL, according to the treatments under study and their controls (see Table 1). Then, it conforms to each experimental unit. Each experimental treatment, as well as the positive or negative controls, was carried out in triplicate.

First, two solutions of 10 mL total volume were prepared in a volumetric flask of the same volume. In the first, 20 µL of adjuvant and 5 mL of deionized water were added. After mixing the first two components, 10 µL of the commercial insecticide was added and graduated to 10 mL with deionized water. In the second, only 20 µL of adjuvant and about 5 mL of deionized water were mixed for a few seconds, and then the remaining deionized water was added until it reached 10 mL.

Cigaral 35 SC®, a chemical insecticide based on a concentrated suspension of imidacloprid ($C_9H_{10}ClN_5O_2$, CAS Number: 138261-41-3) at 350 g·L$^{-1}$, was used as a commercial insecticide.

On the other hand, COSMO IN®-d, a non-ionic adjuvant based on ethoxylated alcohol and polyoxyethylene alkyl ether, was used as an adjuvant.

With the first solution ("Sol. A"), the "positive control" of the experiment was elaborated, where commercial insecticide was used. With the second ("Sol. B"), the solution was prepared in which the essential oil extracts from the tangerine peel obtained were diluted, and the "negative control" of the experiment was formed (Table 1).

Once the experiment started, the dying whiteflies were counted every 3 h. The cumulative mortality was calculated as follows:

$$\text{Cumulative mortality (\%)} = \frac{\text{Cumulative dead of adult whiteflies}}{\text{Initial number of adult whiteflies}} \times 100 \qquad (1)$$

## Optical rotation characterization of EO extracts from tangerine peel

The optical rotation of an extract is the angle through which the plane of polarization is rotated when polarized light passes through it. Optically active substances are termed dextro-rotatory (*d*- or "+") when they turn plane-polarized light to the right, while levo-rotatory (*l*- or "-") substances rotate it to the left.

The determination of the optical rotation of EO extracts was carried out with a digital polarimeter (AP-300; ATAGOTM, Bellevue, WA, USA).

## FTIR analysis of EO extracts from tangerine peel

EO extracts from tangerine peels obtained using solvent extraction were analyzed by IR spectrometry using an Agilent Cary 630 FTIR (Agilent Technologies Inc., Santa Clara, CA, USA) in a wavenumber range between 400 and 4,000 cm$^{-1}$ over 32 scans with a resolution of 4 cm$^{-1}$. Moreover, a single rebound diamond crystal was sampled using an ATR sampling technique.

**Table 1 Different treatments were used in each experimental block.**

| Treatment | Preparation |
|---|---|
| "Positive control" (C⁺) | 0 µl EOE[1] + 177.8 µL of Sol. A[2] |
| T1—12.5% (*v/v*) | 22.2 µL EOE + 156.6 µL of Sol. B[3] |
| T2—25.0% (*v/v*) | 44.4 µL EOE + 133.4 µL of Sol. B |
| T3—33.3% (*v/v*) | 59.3 µL EOE + 118.5 µL of Sol. B |
| "Negative control" (C⁻) | 0 µL EOE + 177.8 µL of Sol. B |

Notes:
[1] EOE: essential oil extract.
[2] Sol. A (solution of 0.1 % (v/v) imidacloprid + 0.2 % (v/v) COSMOS IN®-d).
[3] Sol. B (solution of 0.2 % (v/v) COSMOS IN®-d).

## HPLC Analysis of EO extracts from tangerine peel

High-Performance Liquid Chromatography (HPLC) apparatus Ultimate 3000, equipped with reverse phase C-18 column for HPLC Hypersil GOLD™ (150 × 4.6 mm), an autosampler, a Thermo Scientific quaternary pump, (Thermo Fisher Scientific, Waltham, MA, USA), a column compartment, and a photodiode array detector (PAD) was used for the analysis of the samples. The chromatograms were developed at 280 nm of wavelengths, using a linear gradient $H_2O$/MeCN (95:5) to (20:80) over 8 min. The UV-Vis profile for each peak was also obtained with this equipment.

## Gas chromatography and mass spectrometry characterization

APGC-Mass spectrometric detection was carried out on 7890A gas chromatograph equipped with SYNAPT G2 HDMS detector, with an Agilent column DB5-MS, 30 m length, 0.25 mm I.D., 0.25 µm (5% phenyl and 95% polydimethylsiloxane). Split injection 100:1 and injection temperature of 230 °C were used with helium as carrier gas. The temperature gradient was 70 °C, 5 °C/min to 300 °C. The transfer line temperature was 310 °C.

Mass spectrometry was detected on SYNAPT G2 HDMS (Waters Corporation, Milford, MA, USA) using TOF-MSE (Pos. ion mode) and APCI ionization. The corona current and sampling cone voltage were 2.8 µA and 30 V, respectively. A source temperature of 150 °C, a cone gas of 50 L·h⁻¹, and an auxiliary gas flow of 500 L·h⁻¹ were also used.

## Statistical analysis and comparison between treatments

The statistical comparison of the average cumulative mortality for each of the treatments and its controls, both between treatments and for each time that elapsed, was carried out using the open-access statistical R-package, version 4.0.5 (2021-03-31).

# RESULTS

## Cumulative mortality experiments

The results of the experiments on the average mortality rate of greenhouse whiteflies using tangerine oil extracts and petroleum ether (PET) and n-hexane (HEX) solvents were very similar (Fig. 1).

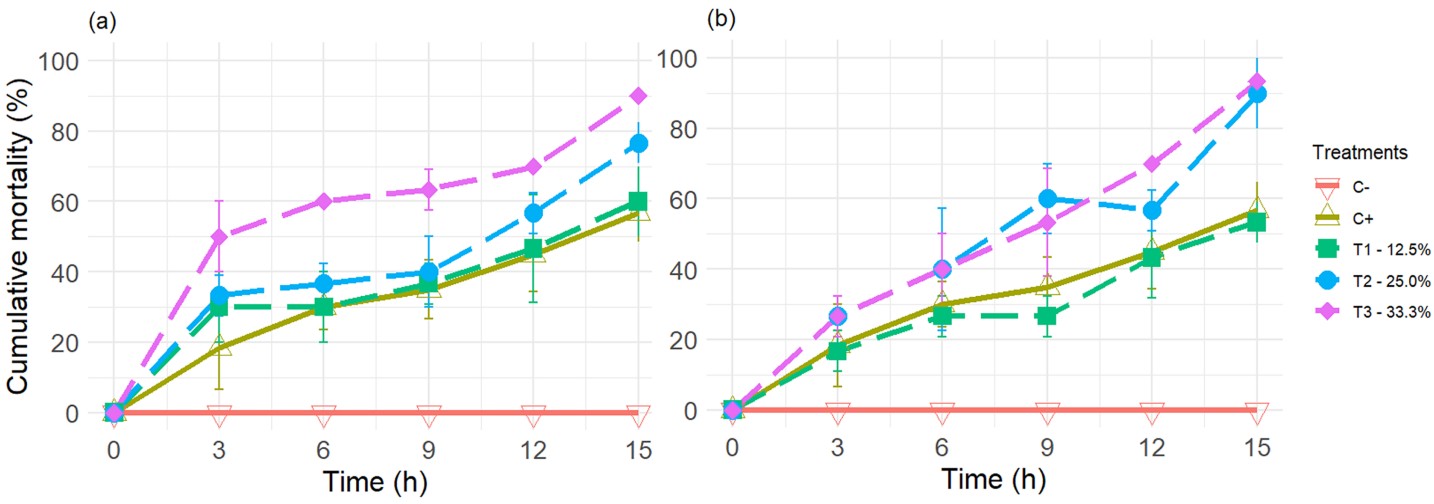

**Figure 1 The cumulative mortality of adult greenhouse whiteflies was placed in contact with the vapours of controls (positive and negative) and the three treatments used.** Treatments: T1—12.5%, T2—25.0%, and T3—33.3% (v/v) of extracts of essential oils (EOEs) (see Table 1 for details), obtained from tangerine peels, using as solvents (A) Petroleum ether (PET), and (B) n-hexane (HEX). Each point and bar represent the average ± standard deviation values ($n = 3$).

However, the cumulative mortality values ($n = 180$) did not follow a normal distribution, according to the Shapiro-Wilks test performed.

Additionally, the presence of the EO components of the tangerine peel and imidacloprid, the active ingredient in the commercial insecticide, had a lethal effect on the greenhouse whitefly during the experiment's duration.

As evidenced by the fact that all treatments plus the positive control (C+) produced significantly different and higher values than the negative control, when the treatments were compared among themselves and with C+ for each of the extracts obtained with PET and HEX, employing a Wilcoxon-Mann-Whitney non-parametric paired rank sum test, it was found that there were no significant differences ($p < 0.05$) between treatments T1 to T3 and between these and C+, for EO extracts obtained with HEX.

In contrast, for the EO extracts obtained from PET, significant differences were observed between T3 and T1, T2, and C+, but not between T1, T2, and C+ among themselves ($p < 0.05$) (see Supplemental Material annexed).

## Optical rotation characterization of EO extracts from tangerine peel

For the EO extracts analyzed, the optical rotation is dextro-rotatory with an angle between +95° and +97° for both EO extracts (with n-hexane and petroleum ether), suggesting the presence of *d*-limonene.

## FTIR-characterization of EO extracts from tangerine peels

The FTIR spectroscopy analyses of the final samples of the EO extracts of tangerine (*Citrus reticulata* L.) peels obtained by extraction with PET and HEX show remarkable similarities (Fig. 2).

The peaks shown in the FTIR spectra represent the main interactions between the atoms present mainly in *d*-limonene. Thus, for example, the broad peak between 3,100 and

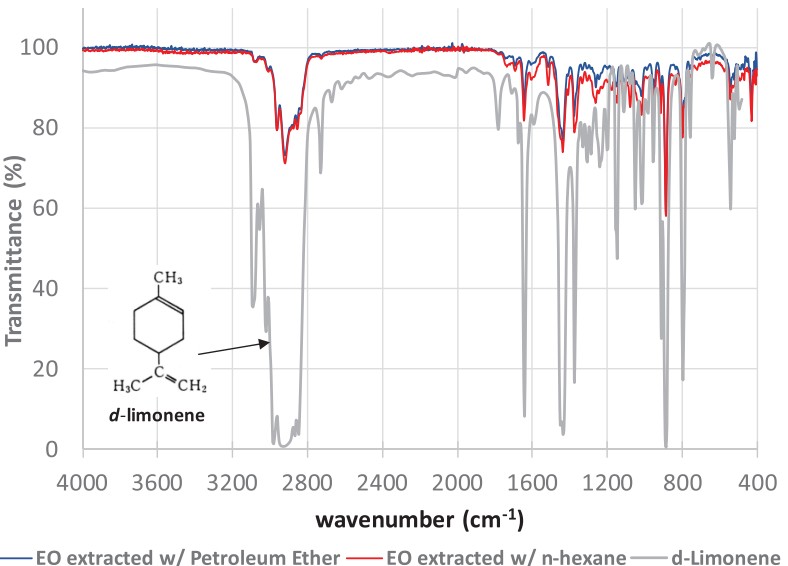

**Figure 2** FTIR spectra of the essential oil extracts of tangerine peels extracted with Petroleum Ether (blue line) and n-hexane (red line) compared with the reported FTIR spectrum of d-limonene (https://webbook.nist.gov/cgi/cbook.cgi?ID=C5989275&Mask=80#IR-Spec).

2,800 cm$^{-1}$ represents the asymmetric stretching of the C-H bond in the methyl (-CH$_3$) and methylene (-CH$_2$) groups. The broad peak between 1,600 and 1,620 cm$^{-1}$ defines the C = C bond stretching in the cyclohexene ring. Peaks at approximately 1,450 cm$^{-1}$ describe the doublet stretching of the C-H bond in the methylene groups. Whilst peaks at around 1,160 and 990 cm$^{-1}$ represent the C-C bond in the methylene and methyl groups and the cyclohexene ring present in the molecule, respectively.

### GC-MS characterization of EO extracts from tangerine peels

The HPLC profile showed three majority peaks at 1.6, 2.1 and 3.6 min with 17, 26 and 41%, respectively (Fig. 3A). Considering the UV profile, the peak at 3.6 min was collected, and lyophilized. Further, the samples were analyzed by GC-(TOF) MS allowed to confirm the *d*-limonene presence with a m/z of 136.0509 (Fig. 3B).

Gas chromatography coupled with mass spectrometry is an exact and versatile technique for analyzing volatile compounds such as limonene. This method combines the separation of mixtures by gas chromatography with precise identification and through mass spectrometry. Samples of the EO extracts obtained with n-hexane and petroleum ether extractions confirmed the presence of *d*-limonene, not only for its unmistakable characteristic odor but also corroborated, in both cases, by its optical rotation values and their corresponding molecular mass.

The experiments were conducted on a TOF-MS system with continuous interleaving scans at low and high collision energies (MSE). Specifically, the MSE mode allowed for the simultaneous acquisition of full-spectrum accurate mass data of both parent and fragment ions in a single chromatographic run. The mass spectra of all limonene samples showed a majority peak (parent/molecular ion) corresponding to limonene mass m/z 136.0509,

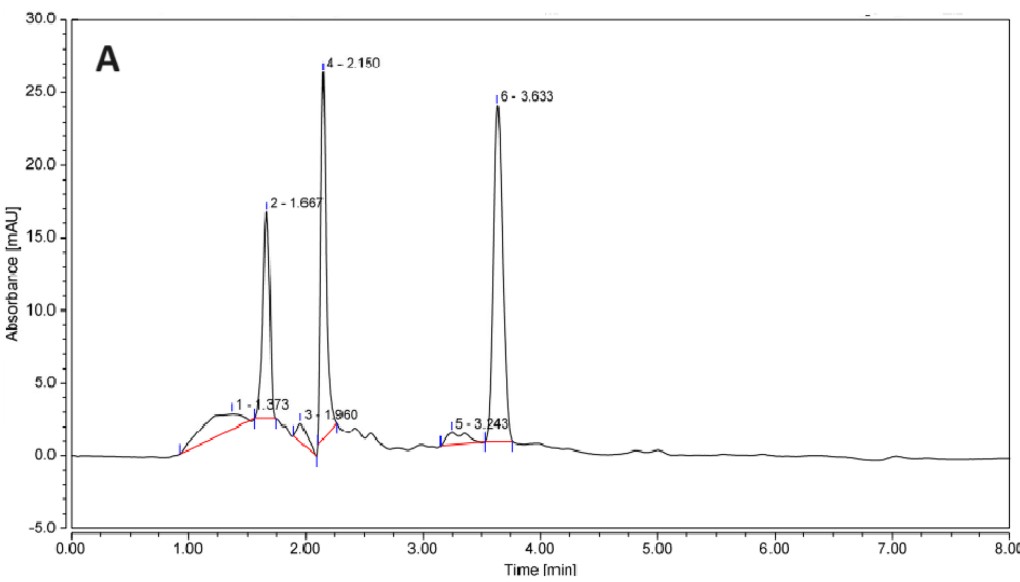

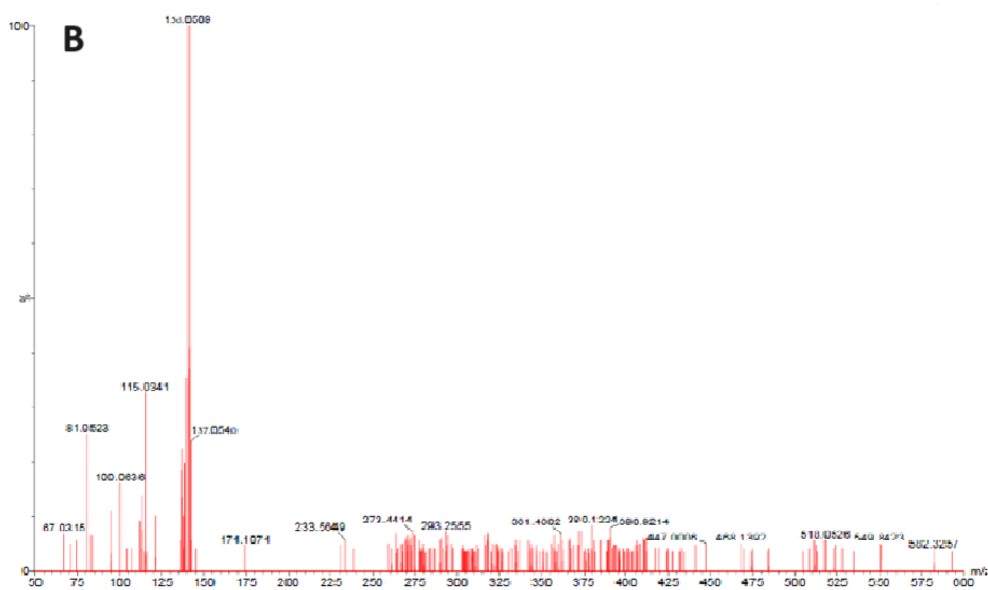

| Peak No. | Retention Time, min | Area, mAU*min | Height, mAU | Relative Area, % | Relative Height, % |
|---|---|---|---|---|---|
| 1 | 1.373 | 0.540 | 1.055 | 10.33 | 1.61 |
| 2 | 1.667 | 0.900 | 14.179 | 17.24 | 21.57 |
| 3 | 1.960 | 0.131 | 1.304 | 2.51 | 1.98 |
| 4 | 2.150 | 1.353 | 25.203 | 25.92 | 38.35 |
| 5 | 3.243 | 0.157 | 0.864 | 3.01 | 1.31 |
| 6 | 3.633 | 2.141 | 23.122 | 41.00 | 35.18 |

**Figure 3 Characterization of the essential oil of mandarin peel extracted with n-hexane.** (A) Chromatogram profile by HPLC-RP. (B) GC-MS spectrum of peak #6.

corroborating limonene's presence and purity, based on accurate mass, MSE data and related literature.

Additionally, neither n-hexane (MW 86.18 g·mol$^{-1}$) nor petroleum ether (MW 82.20 g·mol$^{-1}$, result not shown) was observed in any of the mass spectra.

## DISCUSSION

Some studies have reported some citrus EO extracts' larvicidal and insecticidal activity against different pests. Thus, for example, larvicidal activity has been registered against *Bactrocera tryoni* (*Muthuthantri et al., 2015*), *Aedes aegypti* (*Bailão et al., 2022*), and *Culex pipiens* (*Azmy et al., 2021*).

However, there are few reports on using EO extract from *Citrus reticulata* L. peels to control greenhouse whiteflies. Only the use of lemon (*Citrus aurantifolia* H.) peels EO extract has been reported, with observations of insecticidal activity at all stages of the life cycle of the species of *Trialeurodes vaporariorum* W. (egg, nymph, and adult stages) (*Delkhoon et al., 2013*).

On the other hand, other aromatic plant extracts with monoterpenes, such as d-limonene, have been reported for their insecticidal activity. In this way, for example, the repellent and anti-oviposition activities of five aqueous plant extracts (*Foeniculum vulgare* (seed), *Achillea millefolium* L. (leaves), *Cuminum cyminum* L. (seed), *Thymus vulgaris* L. (leaves and flowers), and *Citrus sinensis* L. (peel)) on adult greenhouse whiteflies are suggested. The presence of monoterpenes in the EO extracts is indicated in all cases. In this study, however, the lowest repellent and oviposition effects were observed in *C. sinensis* peels, which could suggest that water is not a suitable extraction solvent to extract the active principles (*Dehghani & Ahmadi, 2013*).

Although the exact mechanism by which d-limonene kills greenhouse whitefly is not entirely known (*Ibáñez, Sanchez-Ballester & Blázquez, 2020*), it may involve several different modes of action, which could favor its long-term use in the pest control of insects in greenhouses without promoting the appearance of resistance phenomena.

Some previous studies suggest that the main mechanism is to repel, without killing the whitefly by the action of a volatile compound such as *d*-limonene (*Conboy et al., 2020*). However, it is possible that in closed recipient, such as in the experiment carried out here, to the repellent action, some other toxic effect is added, which would kill the adult whiteflies. An observation points to the diffusion of this volatile compound to the inside of the whitefly and the change in the coloration of the dead whiteflies, changed from white yellowish for the living whiteflies to a dark-brown color for the dead whiteflies.

The extraction of essential oils and d-limonene from citrus peels has been achieved with different yields using different methods and techniques. Hydro-distillation by stripping with steam (*Ramgopal et al., 2016*), the use of the Soxhlet apparatus and organic solvents such as petroleum ether, n-hexane, and ethyl alcohol (*Park, Ko & Kim, 2015*; *Mirdha & Routray, 2020*), the use of supercritical fluid of $CO_2$ (*Filho et al., 2003*), as well as extractions assisted by microwaves (*Auta et al., 2018*) and ultrasound (*Khandare, Tomke & Rathod, 2021*).

The yields obtained in this study (1.6 and 2.0 % (m/m) for PET and HEX, respectively) are close to those reported by other authors, who achieved d-limonene yields of 3 and 5% (m/m) using PET and HEX as solvents (*Park, Ko & Kim, 2015*).

Interestingly, some authors report better extraction yields for d-limonene, using ethyl alcohol as an extraction solvent, whose polarity is greater than that of n-hexane and

petroleum ether (*Park, Ko & Kim, 2015*; *Mirdha & Routray, 2020*). For example, a d-limonene yield of 78 % was achieved using ethyl alcohol as a solvent, a significantly higher yield than those reported using the previous mentioned non-polar solvents (*Park, Ko & Kim, 2015*).

The difference that has been observed with ethanolic extraction is that the color of the ethanolic extract is dark brown instead of the typical yellow hue of the n-hexane extract, which could indicate that together with terpenes, such as *d*-limonene, other components are being extracted with the ethanol, such as pectin, which is also abundant in the tangerine peel (J Prado and M Cañarejo, 2023, personal observation).

The characteristic citrus odor mentioned by several authors (*Attard et al., 2014*; *Keskin, 2020*) is commonly associated with the presence of the *d*-limonene isomer. The positive polarization values near to +96° were obtained for a sample of tangerine peel essential oil extracted with n-hexane and petroleum ether, respectively, corroborating the presence of *d*-isomer. These values were similar to that those previously reported (*Javed et al., 2014*; *García-Fajardo et al., 2023*).

Additionally, the FTIR spectra of the tangerine peel essential oil extracts obtained here strongly suggest the presence of *d*-limonene.

It was finally verified by GC-MS analysis of the EO samples, which showed that the molecular mass of the isotopes presents corresponded to that of *d*-limonene. This result agrees with other similar studies, in which EO extract from *C. reticulata* peels was 88.9 % (*Droby et al., 2008*), whilst another study pointed to *d*-limonene, α-farnesene, and β-elemene as the fundamental components of EO extract (*Dong, Shao & Liang, 2014*).

One of the possible reasons given by various authors (*Alotaibi et al., 2022*) for observing the relatively low levels of resistance of botanical or natural insecticides compared to conventional chemical insecticides is the presence of other organic compounds in addition to *d*-limonene. This may make it difficult for insects' natural defense mechanisms to make the necessary "adjustments" to their DNA code to quickly adapt to the presence of a chemical that is toxic to them.

## CONCLUSIONS

The tangerine peel, a by-product of the tangerine agroindustry, is an excellent example of the valorization of agricultural waste (*Panwar, Panesar & Chopra, 2021*; *Sharma et al., 2022*). Obtaining an essential oil from the peel through extraction with organic solvents and its subsequent removal by distillation provides a product of natural origin that has very low phytotoxicity in potatoes (*Solanum tuberosum*) (J Prado and M Cañarejo, 2023, personal observation), which has activity as a botanical insecticide (*de Brito et al., 2021*).

Despite its lower availability than traditional chemical insecticides (*Ahmed et al., 2022*), a natural bio-insecticide based on extracting essential oil from the tangerine peel and used in closed environments such as greenhouse crops could provide benefits for controlling some insect pests in long-term agricultural practices. Additionally, it could improve the health and working conditions of hundreds of Ecuadorian women workers currently employed in greenhouse flower production.

### Funding

The Dean, Marcelo Cevallos of the FICAYA Faculty and Rector, Miguel Naranjo Toro, organized and approved funds from the Central Government to Pay for this Article. The funders had no role in study design, data collection and analysis, decision to publish, or preparation of the manuscript.

### Grant Disclosures

The following grant information was disclosed by the authors:
FICAYA Faculty and Rector.
Central Government to Pay for this Article.

### Competing Interests

The authors declare that they have no competing interests.

### Author Contributions

- Nancy Flores performed the experiments, prepared figures and/or tables, and approved the final draft.
- Julia Prado conceived and designed the experiments, analyzed the data, authored or reviewed drafts of the article, and approved the final draft.
- Rosario Espin analyzed the data, prepared figures and/or tables, and approved the final draft.
- Hortensia Rodríguez performed the experiments, prepared figures and/or tables, and approved the final draft.
- José-Manuel Pais-Chanfrau conceived and designed the experiments, analyzed the data, prepared figures and/or tables, authored or reviewed drafts of the article, and approved the final draft.

### Data Availability

The raw data are available in the Supplemental Files.

### Supplemental Information

Supplemental information for this article can be found online at http://dx.doi.org/10.7717/peerj.16885#supplemental-information.

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
