# Peer review of "Laboratory evaluation of a bio-insecticide candidate from tangerine peel extracts against Trialeurodes vaporariorum (Homoptera: Aleyrodidae)"

_PeerJ, doi:10.7717/peerj.16885_

## Round 0.1 · original submission · Major Revisions

Modify manuscript according to the reviewers' comments.

Reviewer 1 ·

Basic reporting

The article mentions the use of the evaluation of a bio-insecticide candidate from tangerine peel extracts against Trialeurodes vaporariorum.
English used in the manuscript was very professional.
The raw data, Tables, and figures which are shared are relevant to the findings. Some more details need to be incorporated.
Literature references were relevant but need there are some corrections to be done.

Experimental design

The research article clearly demonstrates the use of limonene extract in the control of white flies.
Though a lot of research on limonene has been published and it has already been used in many different areas, including FMCG, Pharmaceuticals, cancer studies, as an insecticide, in controlling bacterial infections, etc. The experimental section and methods are well explained.

A lot of papers are there which give more efficient extraction procedures.

Since the paper mentions d-limonene - more data is required which proves that it is d-limonene. FTIR data is not sufficient to characterize the chirality of the molecule. Kindly refer to Khan et al., 2010 and - Ibáñez MD - 2020 for more details. Also, need to add GCMS analysis for checking the % of the different components present in the Tangerine peel extract.

Validity of the findings

Need more data like GCMS and chiral analysis details which confirm it's a d-limonene.
Since there is a lot of work already carried out, this research focuses on the use of limonene on white flies.

The conclusion is well stated.

Additional comments

• Line-375 - Ibáñez MD, Sanchez-Ballester NM, Blázquez MA. 2020. Encapsulated limonene: A pleasant lemon-like aroma with promising application in the agri-food industry. A review. Molecules 25. DOI: 10.3390/molecules25112598. – d and l - Limonene
• Line-393 - Khan I, Shah ZA, Saeed M, Shah HU. 2010. Physicochemical analysis of Citrus sinensis, Citrus reticulata and Citrus paradisi. Journal of the Chemical Society of Pakistan 32.


• Line-96, 413 - Mandal S, Mandal M. 2016. Tangerine (Citrus reticulata L. var.) Oils. Essential Oils in Food Preservation, Flavor and Safety:803–811. DOI: 10.1016/B978-0-12-416641-7.00091-2. Kindly recheck and share the actual paper details.
• Line-138,441 - Park SM, Ko KY, Kim IH. 2015. Optimization of d-limonene Extraction from Tangerine Peel in Various Solvents by Using Soxhlet Extractor. Korean Chemical Engineering Research 53. DOI: 10.9713/kcer.2015.53.6.717. – Pl. share actual paper details

• This research activity is already published and available online.
Bio-Insecticidal Potential of Tangerine Peel Oil Against Greenhouse Whitefly: A Green Biopesticide Candidate Nancy A. Flores-Mediavilla 1, Julia K. Prado-Beltrán 2, Rosario del C. Espín-Valladares 1, Hortensia María Rodríguez-Cabrera 3 and José Manuel Pais-Chanfrau 1* 1 Carrera de Agroindustria, FICAYA, Universidad Técnica del Norte (UTN), Ave. 17 de Julio 5-21 & José María de Córdova, Ibarra 100115, Ecuador; [email protected] (N.A.F.-M.); [email protected] (R.d.C.E.-V.). 2 Carrera de Agropecuaria, FICAYA, Universidad Técnica del Norte (UTN), Ave. 17 de Julio 5-21 & José María de Córdova, Ibarra 100115, Ecuador; [email protected] (J.K.P.-B.) 3 School of Chemical Sciences and Engineering, Yachay Tech University, Hacienda San José s/n y Proyecto Yachay, Urcuqui 100119, Ecuador; [email protected] (H.M.R.-C.) * Corre

Annotated reviews are not available for download in order to protect the identity of reviewers who chose to remain anonymous.

·

Basic reporting

The introduction of the manuscript requires improvement in terms of precision and clarity. The current version is challenging to follow, and I suggest focusing on refining the English grammar. The sentences are too long, resulting in ambiguity in certain instances. To improve readability and comprehension, I recommend rewriting the introduction to provide a more concise and coherent overview of the study. Addressing these issues will significantly improve the overall quality of the manuscript and ensure that readers can quickly grasp the main points and objectives of the research.

I would recommend improving the quality of the figures. Firstly, "Figure 1" appears to be quite blurred. I suggest enhancing the image resolution to ensure clarity. Additionally, "Figure 2" seems suitable for inclusion in the supplementary material rather than the main text, as it may not be directly relevant to the primary findings. Furthermore, "Figure 3 B" appears unnecessary and could be omitted altogether. Lastly, I recommend revising the legend of "Figure 4" to provide a more explicit description of the treatment groups, allowing readers to understand the experimental setup better.

Experimental design

The methods section of the manuscript requires significant improvement. The "solvent extraction at laboratory scale" description is unclear and difficult to comprehend. I strongly recommend re-visiting the English grammar throughout the manuscript to enhance clarity. In addition, the presentation of the methods should be simplified to avoid confusion. Clear and concise descriptions are essential for the reader to understand the experimental procedures. Additionally, the statistical tests employed in the study need to be explained more clearly. Currently, they are not understandable, and readers may struggle to grasp the analytical approach. I suggest providing more detailed explanations of the methods and statistical tests used to enhance the overall clarity of the methodology.

Validity of the findings

The manuscript's results section lacks clarity and could be improved for better comprehension. I recommend presenting the results simply by indicating the percentage of dead whiteflies in each treatment. This will facilitate a clearer understanding of the outcomes. Additionally, it would be helpful to move the statistical tests used to the methods section, as their inclusion in the results section adds confusion and hampers clarity. The discussion section appears to be lacking in depth. It would have been beneficial for the authors to expand the discussion to include other compounds found in citrus peels, such as farnesenes and their potential contribution to the insecticidal properties. This would provide a more comprehensive understanding of the chemical composition and efficacy of the essential oil of the tangerine peel as an insecticide.

Additional comments

Line 46: yield of what?
Line 50: chemicals such as pesticides? I would suggest being precise.
Lines 51-52: What are the adverse effects and what problems? Please give examples.
Lines above 72: English should be corrected. I would encourage authors to phrase the sentences in a simple way and not begin the sentence with words such as "they, this, etc."
Line 76: Various pathologies such as XYZ?
Lines 76-78: Please rewrite the sentence. For example: Furthermore, in agriculture, the prevention of pest damage is crucial during the cultivation of flowers, fruits, and vegetables, as well as during the transportation and storage of agricultural products.
Line 80: Why?
Line 79-85: This paragraph is all over the place. Please give examples and be specific.
Line 88: "…relatively large mandarin plantations". Please be precise. Please give data, some numbers.
Lines 93-96: Using the peel that would go to waste is very good. But what about the main fruit?
Line 100: What is the rationale for this sentence?
Line 129: Maybe please add the coordinates?
Line 137: Was the peel dried before extraction? Please mention it. Also, please add why hexane was used.
Line 146: This sentence is not understandable at all.
Line 153: Where did you buy these boxes? Are they self-made? Please mention the seller.
Line 157: I would suggest dividing the experimental setup into two parts: Greenhouse (large experiment) and Cage (small experiment)
Line 160: Where were the whiteflies bred?
Line 161: Is my understanding correct that the whiteflies were reared in the mentioned entomological boxes and later transferred to the greenhouse? I am confused here.
Line 175: Is COSMO IN-d also imidacloprid?
Line 188: What statistical tests were carried out? Please mention it. What was the nature of the data?
Line 201: It is Shapiro–Wilk test. Please change it.
Line 227-230: What were the means of delivering those EO extracts? Please mention it.
Line 244-247: What could be the mechanism of action? Please speculate rationally and write them down.

Reviewer 3 ·

Basic reporting

The use of nature resources for agriculture has become essential to order to avoid or minimise chemical intake that is detrimental to human health.
The author has explored and demonstrated the use of a natural source, tangerine peel essential oil (d-limonene), as a natural insecticide. The manuscript was well written, and the information provided in the table and figures is related to the research article.

Experimental design

The experiments section is discussed in detail and match them with the earlier reported study. The author has performed an extensive literature search and cited various references relative to the use of d-limonene as a bio insecticide.

Line No. 30: Results - The yield of EO was 1.59% and 2.00% (m/m).
The author should provide information of m/m, where m, means mass, milliliter, or meter?

Suggestions: FTIR characterization of EO extracts from tangerine peels
1. The author may have used the d-limonene reference standard to match the FTIR spectra of EO extracts of tangerine.
2. Limonene is having two isomers, d and l. The author has reported d-limonene based on FTIR data, however the chiral isomers (d or l) cannot be identified and characterized using FTIR technique. Polarimetry (optical rotation), chiral Gas chromatography / GCMS, or NMR data were necessary to it.
3. The author could have provided analytical data (Gas Chromatography) showing that the EO extracts of tangerine used in the study is free of Petroleum ether (PET) and n-hexane (HEX) solvents

Validity of the findings

The use of d-limonene as a bio-insecticide is well recognized and has been described in various research papers. Where the author's findings and conclusions are consistent with earlier research.

Additional comments

The research work is a replication of previously reported work in which the author extracts essential oil (EO) from tangerine peels using petroleum ether (PET) and n-hexane (HEX) as solvents.

---

## Round 0.2 · accepted · Accept

I read it carefully and found that the authors modified the manuscript according to the reviewers' comments.

Reviewer 1 ·

Basic reporting

My comments have been addressed adequatly.

Experimental design

My comments have been addressed adequatly.

Validity of the findings

No Comments

Additional comments

No Comments

Reviewer 3 ·

Basic reporting

The use of natural resources for agriculture has become essential in order to avoid or minimize chemical intake that is detrimental to human health.
The author has explored and demonstrated the use of a natural source, tangerine peel essential oil (d-limonene), as a natural insecticide. The author has accepted the feedback and suggestions from reviewers and revised the manuscript.
The manuscript was well written, and the information provided in the table and figures is related to the research article.

Experimental design

The revised experiments section is discussed in detail and matches them with the earlier reported study. The author has performed an extensive literature search and cited various references relative to the use of d-limonene as a bio insecticide.
The author has added the characterization of d-limonene using optical rotation as well as HPLC and GCMS data in resubmitted paper, which provide accuracy and authentication to the research work.

Validity of the findings

The use of d-limonene as a bio-insecticide is well recognized and has been described in various research papers. Where the author's findings and conclusions are consistent with earlier research.

Additional comments

The research work is a replication of previously reported work in which the author extracts essential oil (EO) from tangerine peels using petroleum ether (PET) and n-hexane (HEX) as solvents.

The resubmitted manuscript can provide more details, which could aid in future studies to enhance the health conditions of greenhouse workers.